# DNA damage induced by topoisomerase inhibitors activates SAMHD1 and blocks HIV-1 infection of macrophages

Petra Mlcochova[1,*] (iD), Sarah J Caswell[2], Ian A Taylor[2], Greg J Towers[1] & Ravindra K Gupta[1,3] (iD)

## Abstract

We report that DNA damage induced by topoisomerase inhibitors, including etoposide (ETO), results in a potent block to HIV-1 infection in human monocyte-derived macrophages (MDM). SAMHD1 suppresses viral reverse transcription (RT) through depletion of cellular dNTPs but is naturally switched off by phosphorylation in a subpopulation of MDM found in a G1-like state. We report that SAMHD1 was activated by dephosphorylation following ETO treatment, along with loss of expression of MCM2 and CDK1, and reduction in dNTP levels. Suppression of infection occurred after completion of viral DNA synthesis, at the step of 2LTR circle and provirus formation. The ETO-induced block was completely rescued by depletion of SAMHD1 in MDM. Concordantly, infection by HIV-2 and SIVsm encoding the SAMHD1 antagonist Vpx was insensitive to ETO treatment. The mechanism of DNA damage-induced blockade of HIV-1 infection involved activation of p53, p21, decrease in CDK1 expression, and SAMHD1 dephosphorylation. Therefore, topoisomerase inhibitors regulate SAMHD1 and HIV permissivity at a post-RT step, revealing a mechanism by which the HIV-1 reservoir may be limited by chemotherapeutic drugs.

**Keywords** DNA damage; HIV; integration; macrophage; SAMHD1
**Subject Categories** DNA Replication, Repair & Recombination; Immunology; Microbiology, Virology & Host Pathogen Interaction
**The EMBO Journal (2018) 37: 50–62**

## Introduction

Retroviruses must reverse transcribe their RNA into DNA and integrate nascent viral DNA into the host genome in order to replicate (Skalka & Katz, 2005; Lusic & Siliciano, 2017). Increasing evidence suggests that macrophage infection contributes to the reservoir of infected cells that persist and prevent cure of HIV/AIDS (Alexaki

et al, 2008; Siliciano & Greene, 2011; Watters et al, 2013; Honeycutt et al, 2016). Integration may be recognized as a form of DNA damage, and the host DNA damage response (DDR) (Daniel et al, 1999; Jackson & Bartek, 2009) is critical for "gap repair" during the integration process. Indeed, there is increasing evidence that key DDR proteins are involved in retroviral infection, specifically during integration (Daniel et al, 2003; Ariumi et al, 2005; DeHart et al, 2005; Lau et al, 2005). Furthermore, proteins able to restrict HIV infection, including APOBEC3G and SAMHD1, have been linked to DNA damage responses (Leonard et al, 2013; Roberts et al, 2013; Clifford et al, 2014; Kretschmer et al, 2015).

SAMHD1 is a deoxynucleotide triphosphohydrolase (Goldstone et al, 2011), and the mechanism of HIV restriction is thought to be via depletion of dNTPs to levels that are insufficient for completion of retroviral reverse transcription (RT) (Goldstone et al, 2011; Lahouassa et al, 2012; Schmidt et al, 2015). The activity of SAMHD1 is thought to be regulated by CDK1/2-mediated phosphorylation at amino acid T592 (Cribier et al, 2013; White et al, 2013; Antonucci et al, 2016; Mlcochova et al, 2017). Here, we link DNA damage in primary myeloid cells and cellular SAMHD1 activation, in the absence of a type I interferon response. We show that DNA damage induced by topoisomerase inhibitors activates p53 and p21, leading to SAMHD1 T592 dephosphorylation/activation. Activated SAMHD1 mediates a block to HIV infection, which occurs after the synthesis of full-length HIV DNA. Importantly, the etoposide (ETO)-induced inhibition of HIV-1 can be fully abrogated by SAMHD1 depletion.

## Results

### DNA damage induces a post-reverse transcription block to HIV-1 in macrophages

It has been reported that HIV can induce DNA damage during integration into host DNA (Daniel et al, 2004). Moreover, certain reports suggest that integration can be enhanced by DNA damage induction (Ebina et al, 2012; Koyama et al, 2013). The cellular response to DNA damage also typically involves cell cycle arrest

1 Division of Infection and Immunity, UCL, London, UK
2 Macromolecular Structure Laboratory, The Francis Crick Institute, London, UK
3 Africa Health Research Institute, Durban, KwaZulu Natal, South Africa
*Corresponding author. Tel: +44 20 7679 2000; E-mail: p.mlcochova@ucl.ac.uk

and activation of DNA damage repair (DDR) (Branzei & Foiani, 2008). We reported recently that HDAC inhibitors, which are known to induce cell cycle arrest and/or apoptosis, inhibit HIV-1 infection in monocyte-derived macrophages (MDM) through activation of SAMHD1 (Mlcochova *et al*, 2017).

Here, we investigate the impact of inducing DNA damage on HIV-1 infection in MDM. We employed topoisomerase inhibitors, which block the unwinding of dsDNA leading to inhibition of fundamental biological processes including DNA replication, transcription, DNA repair and chromatin remodelling, by stabilizing the DNA-single or -double breaks (Hsiang *et al*, 1989; Gorczyca *et al*, 1993). To test this, we treated MDM with 5 μM ETO for 18 h to confirm that ETO induced DNA damage, measured by staining for nuclear γH2AX or 53BP1 (Fig 1A) in uninfected cells. Further, we pretreated MDM with etoposide (ETO) or camptothecin (CTH), inhibitors of topoisomerase I and II, respectively. 18 h after treatment with titrations of inhibitors, we infected cells with VSV-G pseudotyped HIV-1 GFP and measured infection 48 h later by enumerating GFP-positive cells. Both topoisomerase inhibitors blocked HIV-1 infection in a dose-dependent manner (Fig 1B). We then used the most effective, non-cytotoxic, concentration of ETO (5 μM) (Fig 1C) to measure its effect on MDM infection by different viruses (Fig 1D). HIV-1 wild-type and capsid mutants N74D and P90A (known to use alternative host cofactors for nuclear translocation) were equally sensitive to topoisomerase inhibitors. However, HIV-2 and SIVsm, which encode Vpx, and degrade SAMHD1, were insensitive to drug treatment. Infection by DNA viruses, adenovirus (AdV) and herpes simplex virus (HSV), and RNA virus Semliki Forest virus (SFV), was also insensitive to topoisomerase inhibition.

We next investigated where in the HIV-1 life cycle ETO acts. We treated MDM with 5 μM ETO for 18 h and infected cells with VSV-G pseudotyped HIV-1 GFP (known to use an endocytic entry route) or wild-type macrophage and CCR5 tropic HIV-1 isolate BaL and determined infection (Fig 1E and F). Both viruses were equally sensitive to the ETO-mediated inhibition of viral infection, suggesting independence of drug sensitivity from viral route of entry. MDM infected with HIV-1 BaL were used for DNA isolation and qPCR to determine efficiency of reverse transcription (Fig 1G), viral 2LTR circle formation (a measure of nuclear entry) (Fig 1H; De Iaco *et al*, 2013) and viral integration by Alu-gag PCR (Fig 1I). Surprisingly, reverse transcription was not affected by ETO treatment (Fig 1G), but we observed a decrease in 2LTR circles (Fig 1H) and viral integration (Fig 1I). We also confirmed that ETO did not affect RT products at increasing MOI (determined as ffu/target cell), even though infection was significantly decreased (Fig 1J and K). Importantly, ETO did not have any effect on RT products over a time-course (Fig 1L). In addition, although dNTP levels are already very low in MDM, we detected a decrease in dTTP and dCTP after ETO treatment (Fig EV1).

## DNA damage-induced HIV-1 block is SAMHD1 dependent in macrophages

We observed that lentiviruses encoding Vpx (HIV-2 and SIVsm), a SAMHD1 antagonist (Hrecka *et al*, 2011; Laguette *et al*, 2011), were insensitive to ETO in MDM suggesting that SAMHD1 might be responsible for the effect of ETO/CTH on HIV-1 infection (Fig 1D). To test this, we treated MDM with 5 μM ETO, infected cells with HIV-1 and assayed phosphorylation of SAMHD1 at T592 by immunoblot (Fig 2A–C). We found that SAMHD1 is phosphorylated in untreated MDM allowing efficient HIV infection, confirming previous data (Mlcochova *et al*, 2017). Addition of ETO

**Figure 1. Etoposide/Camptothecin-induced DNA damage inhibits HIV-1 infection.**

A   Monocyte-derived macrophages (MDM) were treated with 5 μM etoposide (ETO) for 18 h. Cells were stained for γH2AX and 53BP1 nuclear foci, acquired and analysed using the automated cell-imaging system Hermes WiScan and ImageJ. On average, $10^4$ cells were acquired and analysed ($n = 3$, mean ± s.e.m.; *$P$-value ≤ 0.05; paired $t$-test). Scale bars, 10 μm.

B   MDM were treated with increasing concentrations of etoposide (ETO) and camptothecin (CTH) 18 h before infection. Cells were infected with VSV-G pseudotyped HIV-1 GFP in the presence of ETO/CTH, and $10^4$ cells in each experiment were recorded and analysed for GFP expression 48 h post-infection using an automated cell-imaging system Hermes WiScan and ImageJ ($n = 3$, mean ± s.e.m.).

C   MDM were treated with increasing concentrations of ETO and CTH for 66 h. Cells were stained to distinguish between live and dead cells using LIVE/DEAD fixable Dead cell stain protocol. Percentages of live/dead cells were determined using automated cell-imaging system Hermes WiScan and ImageJ. Addition of 20% ethanol for 10 min into cells treated with ETO was used as positive control ($n = 3$, mean ± s.e.m.).

D   MDM were treated with 5 μM ETO for 18 h and infected in the presence of ETO with VSV-G pseudotyped HIV-1 GFP viruses wild-type (wt) HIV-1 capsid mutants (N74D or P90A), HIV-2, SIV sooty mangabey (SIVsm E543), replication competent adenovirus type 5 AdV (AdV), Semliki Forest virus (SFV) and HSV-1. The percentages of infected cells were determined using an automated cell-imaging system Hermes WiScan and ImageJ and normalized to untreated control ~100% ($n = 3$, mean ± s.e.m.; **$P$-value ≤ 0.01; ***$P$-value ≤ 0.001; (ns) non-significant, paired $t$-test). Cells from a representative donor were used for immunoblotting.

E   MDM were treated with 5 μM ETO for 18 h and infected in the presence of ETO with VSV-G pseudotyped HIV-1 GFP. $10^4$ cells were recorded and analysed for GFP expression 48 h post-infection using an automated cell-imaging system Hermes WiScan and ImageJ ($n = 4$, mean ± s.e.m.; ***$P$-value ≤ 0.001, paired $t$-test).

F   MDM were treated with 5 μM ETO for 18 h and infected in the presence of ETO with full-length replication competent macrophage tropic HIV-1 virus BaL. Cells were stained for intracellular p24, and the percentage of infected cells was quantified 48 h post-infection by FACS ($n = 3$, mean ± s.e.m.; ***$P$-value ≤ 0.001, paired $t$-test).

G–I   MDM were treated with 5 μM ETO for 18 h and infected in the presence of ETO with HIV-1 BaL and DNA isolated 18 h post-infection for qPCR ($n = 3$, mean ± s.e.m.; *$P$-value ≤ 0.05; ***$P$-value ≤ 0.001; (ns) non-significant, paired $t$-test). (G) Late viral RT products. AZT: MDM treated with 20 μM AZT, a reverse-transcriptase inhibitor, were used as control. (H) 2LTR circles. (I) Integrated copies of viral DNA, Alu-Gag qPCR.

J   MDM were treated with 5 μM ETO for 18 h and infected with HIV-1 BaL (3, 6, 12 ffu/cell). Cells were stained for intracellular p24, and the percentage of infected cells was quantified 48 h post-infection by FACS ($n = 3$, mean ± s.e.m.; **$P$-value ≤ 0.01; ***$P$-value ≤ 0.001, paired $t$-test).

K   MDM were treated with 5 μM ETO for 18 h, infected with HIV-1 BaL (3, 6, 12 ffu/cell) and DNA isolated 18 h post-infection for qPCR ($n = 3$, mean ± s.e.m.; (ns) non-significant, paired $t$-test).

L   MDM were treated with 5 μM ETO for 18 h, infected with HIV-1 BaL and DNA isolated 4, 6 and 18 h post-infection for qPCR ($n = 3$, mean ± s.e.m.; (ns) non-significant, paired $t$-test).

Source data are available online for this figure.

led to loss of SAMHD1 phosphorylation and reduced HIV-1 infectivity. As expected, the inhibitory effect of ETO on HIV infection was lost after siRNA-mediated SAMHD1 depletion (SAMHD1 KD) (Fig 2A) or treatment of MDM with SIVmac virus-like particles containing Vpx/Vpr (SIV VLP; Figs 2B and EV2), which was confirmed by a dose titration of HIV-1 virus on SAMHD1 KD (Fig EV3A) and SIV VLP-treated cells (Fig EV3B) and by infecting control and SAMHD1 KD cells at MOI achieving equal percentage

of infected cells (Fig EV3C and D). Importantly, the rescue was independent of the SIV Vpr protein, as evidenced by wild-type activity of SIV VLP deleted for Vpr (Figs 2C and EV2) but completely dependent on presence of Vpx. To investigate which step of the virus life cycle was inhibited by ETO, and consequently rescued by SAMHD1 depletion, we infected MDM with HIV-1 BaL with and without co-infection with SIV VLP and isolated DNA 18 h post-infection to measure late RT products (Fig 2D), 2LTR

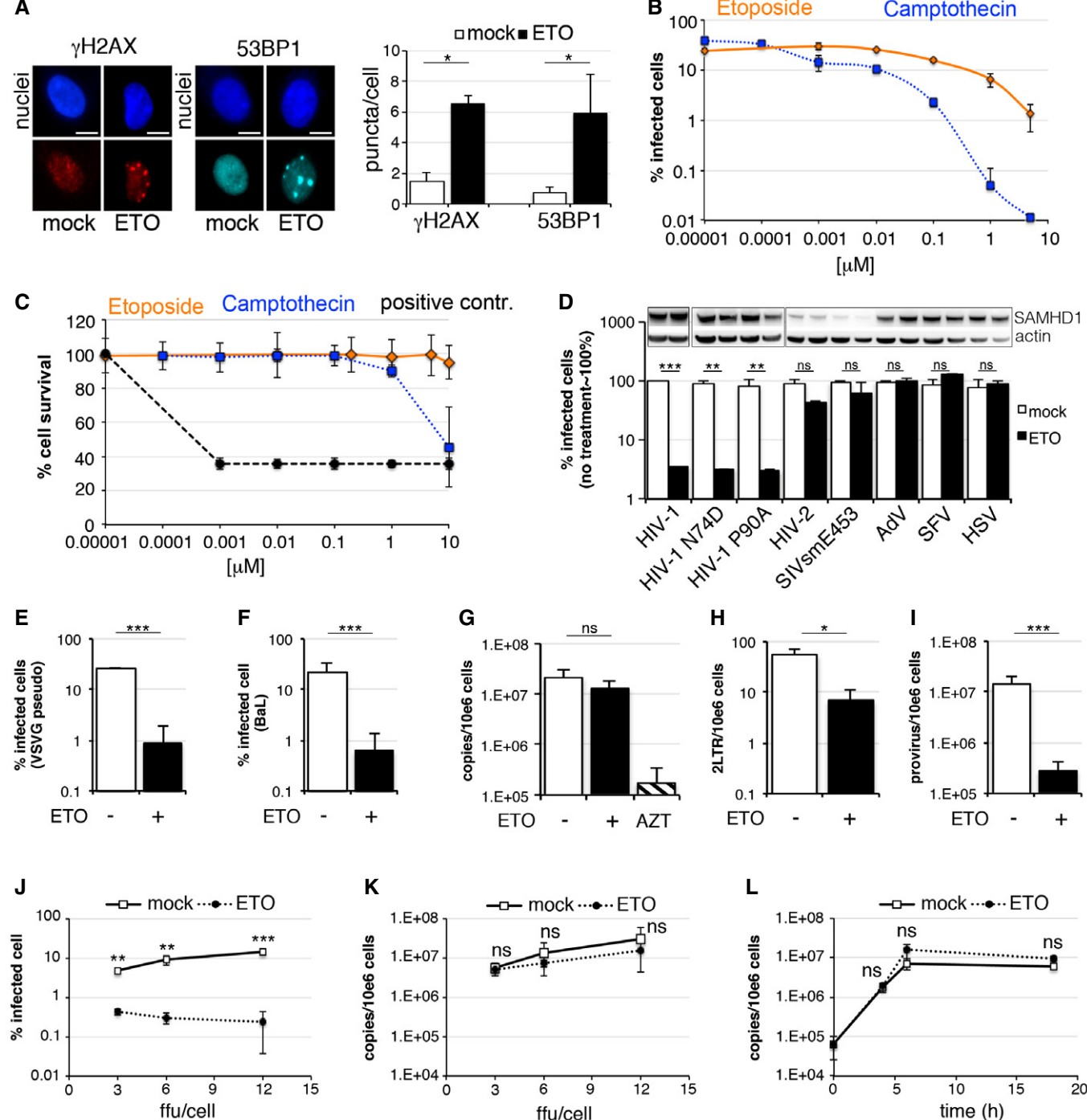

**Figure 1.**

circles (Fig 2E) and integration products (Fig 2F) under conditions of FCS culture where SAMHD1 is phosphorylated in the absence of ETO. Infection was measured in parallel samples 48 h post-viral challenge (Fig 2G). Critically, the post-RT block seen on ETO treatment was abrogated by SAMHD1 depletion. We conclude that HIV inhibition following ETO treatment is mediated through SAMHD1 and impacts both nuclear import and viral integration, but not late viral DNA synthesis.

Surprisingly, we noted that HIV-1 2LTR circles were increased by nearly 20-fold by co-infection with SIV VLP (Fig 2E) in the absence of ETO. However, this increase in 2LTR circles was not mirrored by an increase in integrated proviral DNA (Fig 2E and F).

**SIV bearing Vpx mutant Q76A rescues HIV-1 infection following DNA damage**

To further probe the mechanism of ETO on HIV-1 infection, we treated MDM with 5 μM ETO for 18 h, co-infected cells with HIV-1 and SIV VLP (bearing Vpx WT/Vpr WT) or SIV VLP Q76A (bearing Vpx Q76A mutant/Vpr WT) and measured infection 48 h later. The Vpx mutant Q76A maintains the ability to interact with SAMHD1, but in previous studies did not rescue HIV-1 infection from SAMHD1, possibly because it cannot recruit DCAF1 to degrade SAMHD1 (Srivastava *et al*, 2008; Hrecka *et al*, 2011; Laguette *et al*, 2011; Reinhard *et al*, 2014). We used MDM cultured in human serum (low dNTP levels, dephosphorylated SAMHD1) and in FCS (high dNTP levels, phosphorylated SAMHD1) (Mlcochova *et al*, 2017) and co-infected cells with HIV-1 and SIV VLP or SIV VLP Q76A (Fig 3A and B). SIV VLP but not SIV VLP Q76A increased infection in MDM (Fig 3A and B). By contrast, ETO-mediated inhibition of HIV-1 infection had been rescued by both SIV VLP and SIV VLP Q76A (Fig 3A and B). ETO caused dephosphorylation of SAMHD1 and blocked HIV-1 infection in MDM. However, this block to infection was abrogated by SIV VLP Q76A, even though SAMHD1 was not degraded. As expected, dNTP levels following treatment with SIV VLP Q76A infected cells were not increased (Fig EV4). Critically, SAMHD1 was dephosphorylated (active) in the presence of ETO and SIV VLP Q76A (Fig 3C) suggesting inhibition of the active form of SAMHD1 by the Vpx mutant without manipulation of activation by T592 phosphorylation or degradation.

To confirm which step of the virus life cycle was inhibited by ETO and rescued by SIV VLP Q76A, we infected MDM with HIV-1 BaL with and without co-infection with SIV VLP Q76A and measured infection (Fig 3C), late RT products (Fig 3D), 2LTR circles (Fig 3E) and integration products (Fig 3F). We detected a ~3- to 5-fold decrease in RT, integration of viral products and infection in the presence of Vpx Q76A. This was independent of ETO treatment (Fig 3C–F). Importantly, there was no reduction in reverse transcription but substantial restriction of integrated provirus formation after ETO treatment, which was rescued by co-infection with SIV VLP Q76A (Fig 3A–F). Interestingly, we noted that HIV-1 2LTR circles were not increased by co-infection with SIV VLP Q76A (Fig 3E) in the absence of ETO. These data suggest that following DNA damage, SAMHD1 can be antagonized without being degraded or phosphorylated at T592.

**ETO-induced DNA damage does not trigger type I IFN responses in MDM**

Recent evidence suggests that DNA damage activates the type I interferon system to anti-microbial responses (Brzozek-Racine *et al*, 2011; Hartlova *et al*, 2015). As SAMHD1 was shown to mediate spontaneous expression and release of IFN when mutated (Crow & Manel, 2015) or deleted (Behrendt *et al*, 2013), we investigated the possibility that DNA damage induction could activate a type I IFN response and block HIV-1 infection in MDM.

To test this, we treated MDM with 5 μM ETO for 18 h to induce DNA damage. As a positive control for innate immune triggering, we treated MDM with cGAMP, the product of the activated DNA sensor cGAS (Sun *et al*, 2013; Wu *et al*, 2013) or IFN-β for 18 h. We then isolated RNA from the cells and tested for specific interferon-stimulated gene (ISG) transcripts (Fig 4A). None of the ISG transcripts tested was strongly elevated after ETO exposure of MDM. Moreover, an immunofluorescence/translocation assay detected 5–20% of IRF3-positive nuclei in MDM after treatment with cGAMP and LPS (positive controls), consistent with activation of innate immune responses. However, there was no IRF3 translocation to the nucleus after 2, 6 or 18 h of ETO treatment (Fig 4B and C), suggesting that a type I IFN response is not strongly activated after ETO-induced DNA damage in MDM.

---

**Figure 2. SAMHD1 inhibits HIV-1 at a post-RT step following DNA damage.**

A   MDM were transfected with control or pool of SAMHD1 (KD) siRNA and 3 days later treated with 5 μM ETO and infected in the presence of ETO with VSV-G pseudotyped HIV-1 GFP 18 h later. Cells from a representative donor were used for immunoblotting. The percentage of infected cells was quantified by the automated cell-imaging system Hermes WiScan and ImageJ 48 h post-infection (*n* = 3, mean ± s.e.m.; \*\**P*-value ≤ 0.01; (ns) non-significant, paired *t*-test).

B   MDM were treated with 5 μM ETO for 18 h and co-infected in the presence of ETO with VSV-G HIV-1 GFP and SIVmac virus-like particles containing Vpx/Vpr (SIV VLP). Cells from a representative donor were used for immunoblotting. The percentage of infected cells was quantified by the Hermes WiScan and ImageJ 48 h post-infection (*n* = 3, mean ± s.e.m.; \*\**P*-value ≤ 0.01; (ns) non-significant, paired *t*-test).

C   MDM were treated with 5 μM ETO for 18 h and co-infected in the presence of ETO with VSV-G HIV-1 GFP and SIVmac virus-like particles containing Vpx/Vpr (SIV VLP) or deleted for Vpx (SIV VLP del Vpx) or deleted for Vpr (SIV VLP del Vpr). Cells from a representative donor were used for immunoblotting. The percentage of infected cells was quantified by the automated cell-imaging system Hermes WiScan and ImageJ 48 h post-infection (*n* = 4, mean ± s.e.m.; \*\**P*-value ≤ 0.01; (ns) non-significant, paired *t*-test).

D–F   MDM were treated with 5 μM ETO for 18 h and co-infected in the presence of ETO with HIV-1 BaL and SIVmac virus-like particles containing Vpx/Vpr (SIV VLP). DNA was isolated 18 h post-infection for qPCR quantification of (F) late RT products; (G) 2LTR circles; (H) integrated viral DNA (*n* = 3, mean ± s.e.m.; \**P*-value ≤ 0.05; \*\**P*-value ≤ 0.01; (ns) non-significant, paired *t*-test). (D) Late viral RT products. AZT: MDM treated with 20 μM AZT, a reverse-transcriptase inhibitor, were used as control. (E) 2LTR circles. (F) Integrated copies of viral DNA, Alu-Gag qPCR.

G   The percentage of infected cells was quantified by Hermes WiScan and ImageJ 48 h post-infection (*n* = 3, mean ± s.e.m.; \*\**P*-value ≤ 0.01; (ns) non-significant, paired *t*-test).

Source data are available online for this figure.

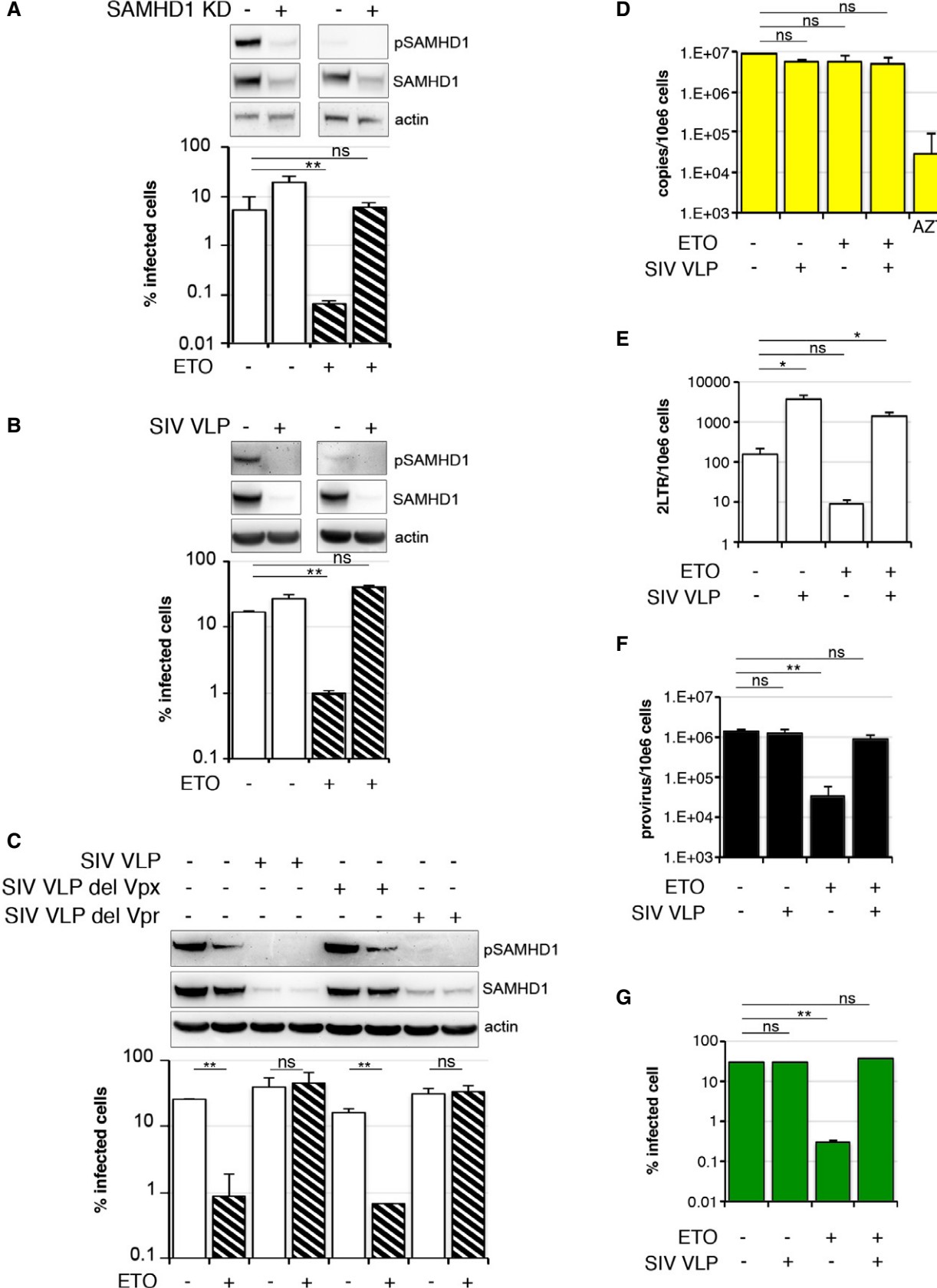

**Figure 2.**

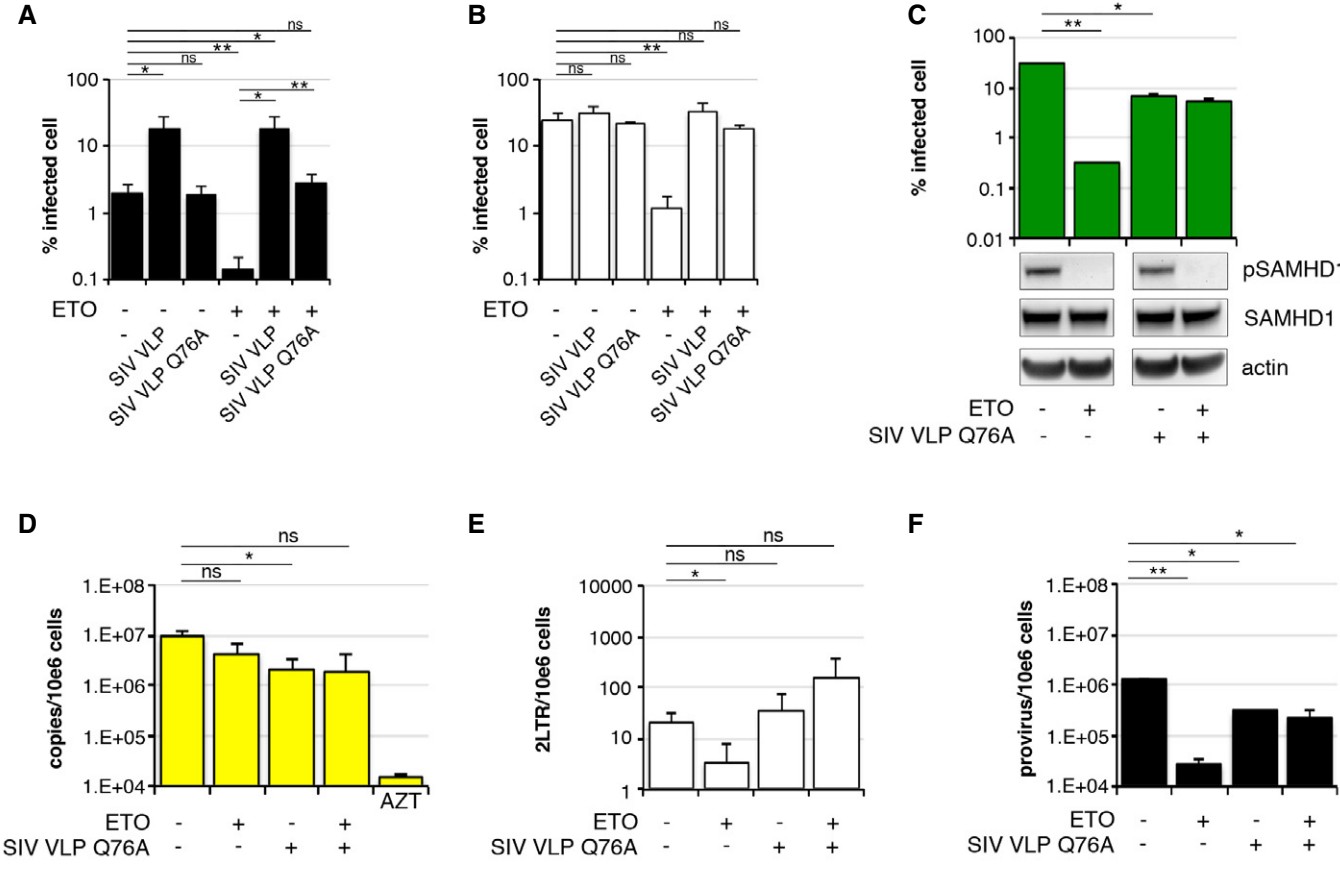

**Figure 3. Vpx Q76A rescues DNA damage-induced block to HIV-1 infection.**

A, B  MDM were treated with 5 μM ETO for 18 h and co-infected in the presence of ETO with VSV-G HIV-1 GFP and SIVmac virus-like particles containing Vpx wild-type (WT)/Vpr or Vpx Q76A mutant/Vpr (SIV VLP Q76A) (n = 3, mean ± s.e.m.; *P-value ≤ 0.05; **P-value ≤ 0.01; (ns) non-significant, paired t-test). (A) MDM were differentiated and cultured in human serum instead of FCS. (B) MDM were differentiated and cultured in FCS. A standard culture condition used in all experiments. See Materials and Methods.

C–F  MDM were treated with 5 μM ETO for 18 h and co-infected in the presence of ETO with HIV-1 BaL and SIV VLP Q76A. DNA was isolated 18 h post-infection for qPCR quantification of (D) late RT products; (E) 2LTR circles; (F) integrated viral DNA (n = 3, mean ± s.e.m.; *P-value ≤ 0.05; **P-value ≤ 0.01; (ns) non-significant, paired t-test). (C) The percentage of infected cells was quantified by the automated cell-imaging system Hermes WiScan and ImageJ 48 h post-infection. Cells from a representative donor were used for immunoblotting. (D) Late viral RT products. AZT: MDM treated with 20 μM AZT, a reverse-transcriptase inhibitor, were used as control. (E) 2LTR circles. (F) Integrated copies of viral DNA, Alu-Gag qPCR.

Source data are available online for this figure.

## DNA damage promotes a G0 state and regulates SAMHD1 phosphorylation via p53 and p21

In the absence of general type I IFN responses after ETO treatment, we hypothesized that the restriction of HIV-1 is mediated entirely by dephosphorylation and activation of SAMHD1. Recent evidence demonstrated that cell cycle status regulates phosphorylation of SAMHD1 (Cribier *et al*, 2013; Badia *et al*, 2016; Mlcochova *et al*, 2017). G0 state macrophages encode active, dephosphorylated SAMHD1, but SAMHD1 is phosphorylated and deactivated by CDK1 in a subpopulation of macrophages in a G1-like state (Mlcochova *et al*, 2017). To test whether ETO altered cell cycle status and therefore activated SAMHD1 through the same pathway, we treated MDM with ETO and CTH and measured the proportion of cells in G0 or G1 by detection of MCM2, a marker expressed throughout the cell cycle but not in

G0. In fact, the proportion of MDM expressing MCM2 and thus being in a G1-like state was significantly reduced after ETO/CTH treatment, indicative of cells returning to G0, a state non-permissive to HIV-1 infection (Fig 5A). We mapped the pathway leading to SAMHD1 dephosphorylation and HIV restriction using immunoblotting (Fig 5B–D). ETO induced DNA damage, as measured by an increase in γH2AX, and resulted in increased expression of p53 and p53 phosphorylation at Ser15 (Fig 5B–D). We also observed increased expression of p21 but not p27 protein. Moreover, absence of PARP cleavage (Fig 5B–D) suggested lack of apoptosis, in addition supported by the lack of cell death measured using cell viability/cell survival analysis (Fig 1C). Loss of HIV-1 permissivity following ETO treatment also correlated with loss of CDK1 and SAMHD1 activation by dephosphorylation. No increase in expression or phosphorylation of CDK2 was detected (Fig 5B and C). The same results were

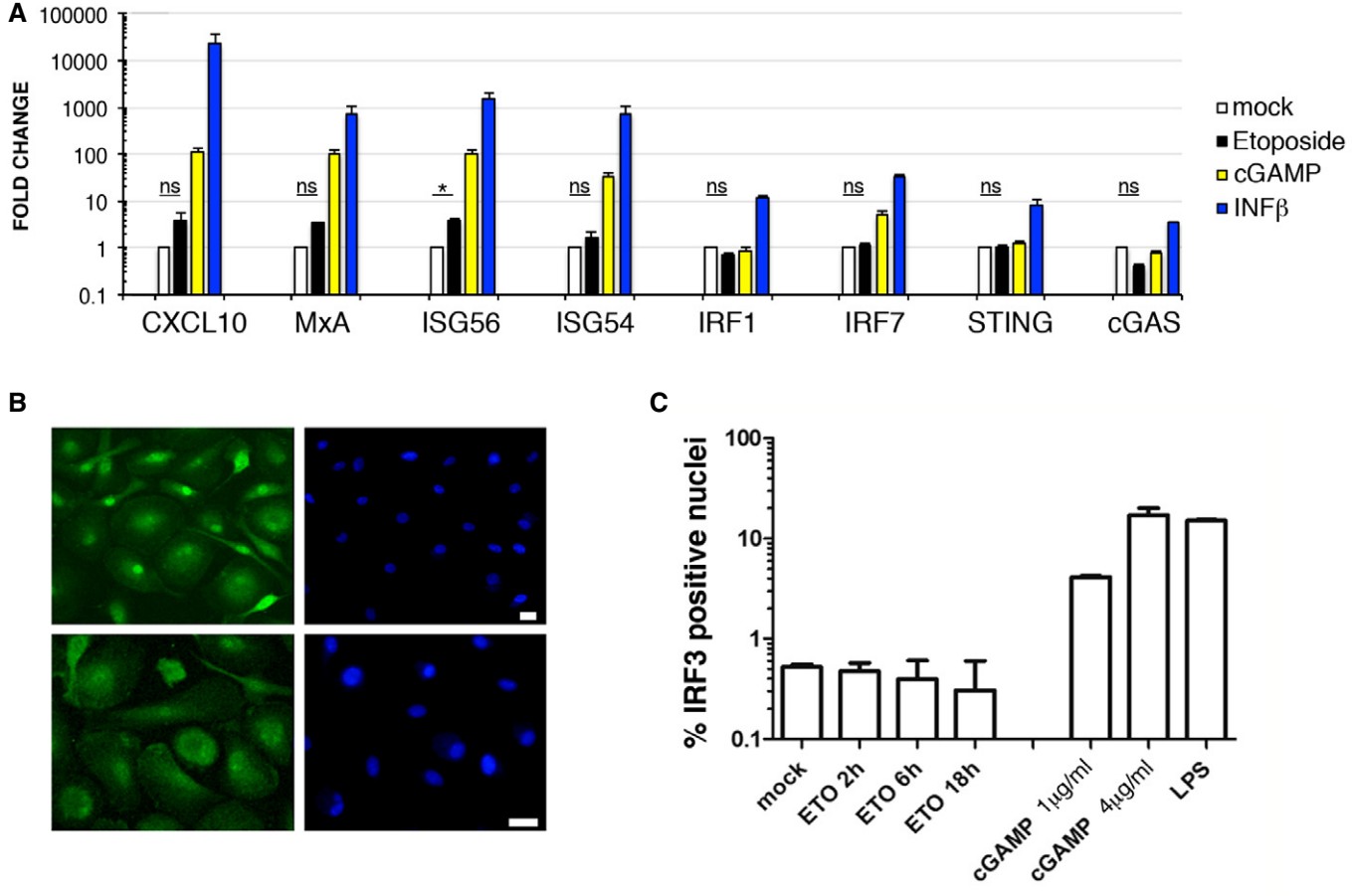

**Figure 4. ETO-induced DNA damage does not activate type I IFN response in MDM.**

A   MDM were treated with 5 μM ETO for 18 h, 3 μg/ml cGAMP and 10 ng/ml IFN-β for 18 h. RNA was isolated and qPCR performed for selected genes using TaqMan assays. Expression levels of target genes were normalized to GAPDH ($n = 3$, mean ± s.e.m.; *P-value ≤ 0.05; (ns) non-significant, paired t-test).

B   MDM were treated with 5 μM ETO or cGAMP for 18 h or 100 ng/ml LPS for 2 h. Cells were stained and analysed for IRF3 translocation (green) into the nucleus (blue). Scale bars: 10 μm.

C   Quantification of nuclei positive for IRF3 staining ($n = 3$, mean ± s.e.m.).

obtained with the topoisomerase I inhibitor CTH (Fig 5D). These data suggest that a DNA damage-induced block to HIV infection in human macrophages is mediated through cell cycle arrest activated by a p53/p21/CDK1 pathway culminating in activation of SAMHD1 by dephosphorylation (Fig 6).

## Discussion

DNA damage and repair have been implicated in regulation of HIV-1 infection, although such reports have opposed each other, likely due to the different cellular backgrounds used (Daniel *et al*, 2003; Ariumi *et al*, 2005; DeHart *et al*, 2005; Lau *et al*, 2005). Indeed, the effect of DNA repair kinases including ATM, ATR, DNA-PK on retroviral infection appears to be variable from cell line to cell line (Yang *et al*, 2010). Here, we present evidence that DNA damage induced by topoisomerase inhibitors used in cancer treatment is associated with SAMHD1 dependent inhibition of HIV-1 infection in primary human MDM. ETO induced activation of SAMHD1 by

dephosphorylation and activated its ability to restrict HIV-1. By contrast, it did not restrict HIV-2 and SIVsm infections because they encode the SAMHD1 antagonist protein Vpx. SAMHD1 was also previously shown to inhibit HSV-1 infection by limiting viral DNA replication, but this inhibition was reported to be independent from SAMHD1 T592 phosphorylation status (Kim *et al*, 2013). Consistent with this observation, we could not detect any effect of ETO on HSV infection in MDM (Fig 1C).

Considerable evidence indicates that SAMHD1 is a dNTP triphosphohydrolase limiting levels of dNTPs and thus retroviral reverse transcription (Goldstone *et al*, 2011; Lahouassa *et al*, 2012; Schmidt *et al*, 2015; Antonucci *et al*, 2016) or herpes virus DNA replication (Kim *et al*, 2013). Nevertheless, other reports have suggested that there might be a second mechanism of SAMHD1 restriction independent of dNTP regulation (White *et al*, 2013; Bhattacharya *et al*, 2016; Welbourn, 2016). Intriguingly, ETO-induced DNA damage was associated with a decrease in dNTP levels, but not with decrease in products of reverse transcription in MDM as might be expected if this mode of SAMHD1-mediated restriction is the result

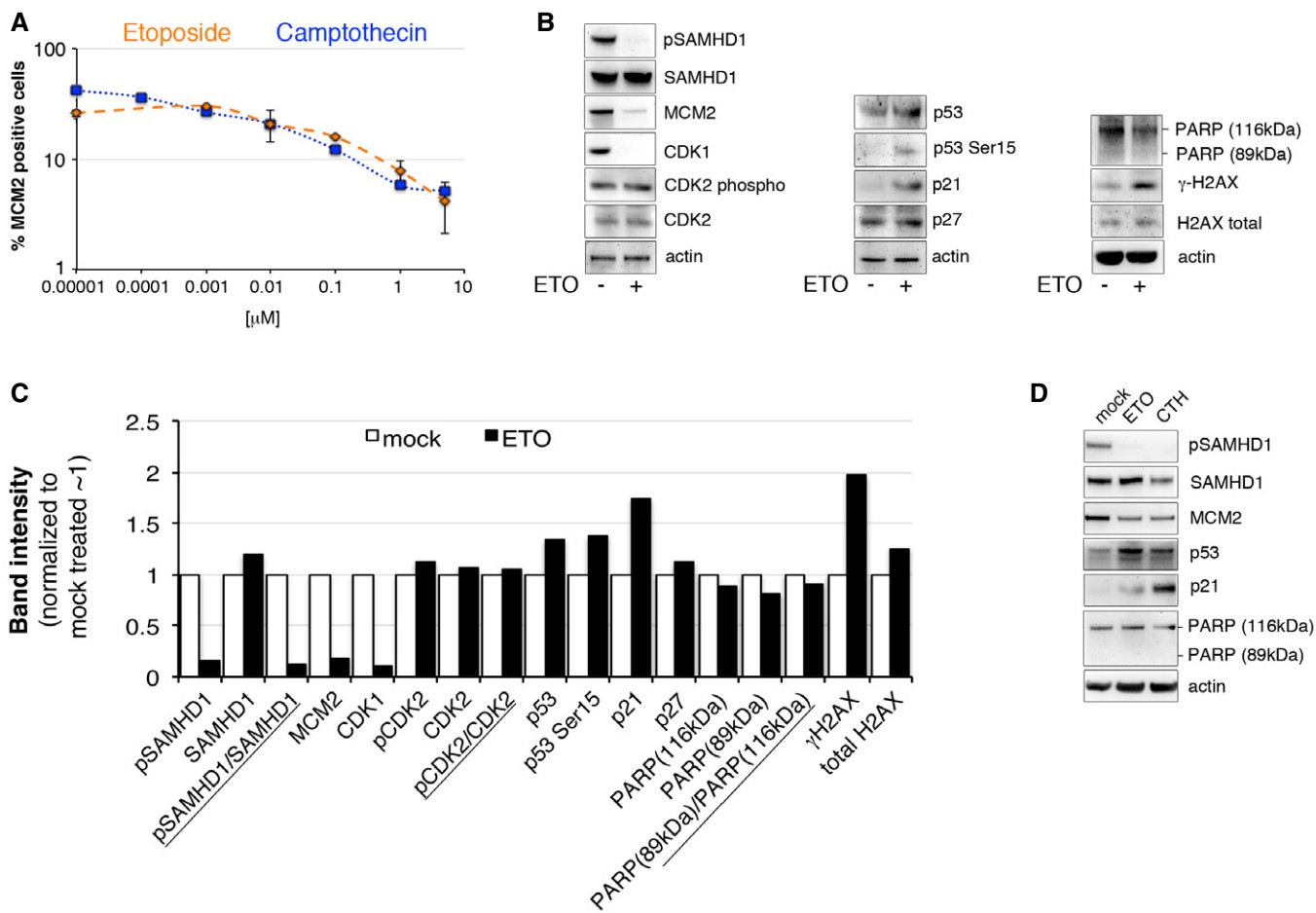

**Figure 5. ETO regulates SAMHD1 phosphorylation through the p53, p21 pathway.**

A  MDM were treated with increasing concentrations of ETO and CTH. Cells were stained for MCM2 expression, acquired and analysed using the automated cell-imaging system Hermes WiScan and ImageJ. On average, $10^4$ cells were acquired ($n$ = 3, mean ± s.e.m.).

B  MDM were treated with 5 μM ETO, lysed and immunoblotting performed to detect cell cycle/cell cycle arrest and DNA damage-associated proteins.

C  Quantification of specific proteins band intensities from immunoblot in panel (B) using a CCD camera. Intensities of protein bands were normalized to intensity of actin protein band.

D  MDM were treated with 5 μM ETO or 0.01 μM CTH for 18 h, lysed and immunoblotting performed to detect cell cycle/cell cycle arrest and DNA damage-associated proteins.

Data information: (B–D) Each panel shows a representative example of three independent experiments.
Source data are available online for this figure.

of regulation of overall cellular dNTP levels and limitation of reverse transcription. Given RT takes place in the cytoplasm, it might be possible that adequate dNTP levels for RT are maintained in the cytoplasm following ETO treatment. Instead, we observed reduced 2LTR circle formation and integrated proviral DNA, indicating a nuclear translocation or integration defect following ETO treatment.

It has been reported that low dNTP levels found in macrophages can restrict DNA gap repair during HIV-1 integration (Van Cor-Hosmer *et al*, 2013) and therefore SAMHD1 could play role in integration through its dNTP triphosphohydrolase activity during this process. Both, formation of 2LTR circles and the gap repair during HIV-1 integration into host genome are recognized by cells as DNA damage and are repaired by host cell DNA damage repair (DDR) machinery (Li *et al*, 2001; Skalka & Katz,

2005). DDR is highly localized into DNA damage nuclear foci (Rothkamm *et al*, 2015) and it might be possible that SAMHD1 alters dNTP concentration locally at DNA damage foci where gap repair takes place. Importantly, SAMHD1 was reported to localize to DNA damage foci in HeLa cells (Clifford *et al*, 2014), and therefore SAMHD1 could play a direct role locally at the site of HIV integration.

The Vpx Q76A variant that is able to interact with SAMHD1, but not DCAF1, fails to cause SAMHD1 degradation (Srivastava *et al*, 2008; Hrecka *et al*, 2011) and fails to increase dNTP levels in dendritic cells (Reinhard *et al*, 2014). Concordantly, we show that Vpx Q76A also fails to increase dNTP levels in MDM. However, we find that this mutant Vpx Q76A is still able to rescue HIV-1 from the effect of ETO even though SAMHD1 protein persists in its active dephosphorylated form. Depletion of SAMHD1 (Fig 2A) illustrates

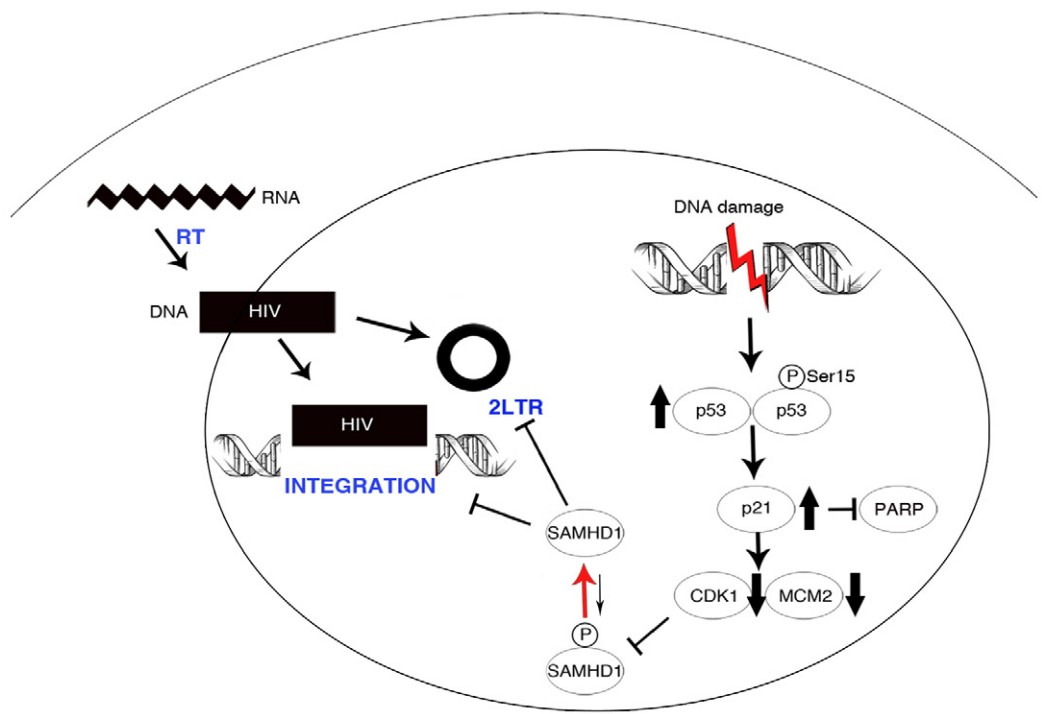

**Figure 6.  Proposed mechanism of action for ETO/CTH and SAMHD1 regulation.**

ETO/CTH-induced DNA damage appears to promote a p53/p21-dependent pathway. This is associated with cell cycle arrest, evidenced by decrease in MCM2 expression. Furthermore, CDK1—responsible for SAMHD1 phosphorylation—is downregulated. As a result, SAMHD1 is not phosphorylated and becomes active against HIV-1 infection. Dephosphorylated, active SAMHD1 induced by DNA damage appears to block 2LTR and provirus formation.

that the effect of ETO on HIV-1 infection in MDM is SAMHD1 dependent. We suggest that Vpx Q76A interaction with SAMHD1 is sufficient to disturb a local SAMHD1 restriction function acting post-RT.

Our experiments with ETO suggest that modulation of dNTP levels can be independent from SAMHD1 restriction, supported by our observation that Vpx Q76A cannot increase dNTP levels but still rescues HIV-1 infection in the presence of ETO. These data are consistent with reports of dNTP independent restriction by SAMHD1 (Bhattacharya *et al*, 2016; Welbourn, 2016). We speculate that SAMHD1 may bind viral DNA directly or indirectly and therefore mediate post-RT restriction. Further investigation is warranted to uncover the mechanism at play.

Intriguingly, co-infection of MDM with HIV and SIV VLP, but not with SIV VLP Q76A, led to an increase in 2LTR circles in untreated cells, reminiscent of observations made in monocyte-derived dendritic cells (Reinhard *et al*, 2014). This observation remains as yet unexplained but suggests that Vpx can distinguish processes leading to 2LTR circle from provirus formation in macrophages.

Recent evidence suggest that DNA damage activates the interferon type I system for anti-microbial responses in murine BMDM (Brzostek-Racine *et al*, 2011; Hartlova *et al*, 2015) or in the human macrophage-like THP-1 cell line (Brzostek-Racine *et al*, 2011). Given that SAMHD1 mutation (Crow & Manel, 2015) or deletion (Behrendt *et al*, 2013) has been associated with spontaneous expression and release of IFN *in vivo* we investigated type I IFN responses in human monocyte-derived MDM after ETO-induced DNA damage. We were unable to detect any large changes to expression of a variety of

interferon-stimulated genes. We conclude that type I IFN responses are not significantly activated after ETO-induced DNA damage in MDM and are therefore not responsible for the block to infection seen.

In the absence of type I IFN related changes we were able to demonstrate that the ETO-induced block is mediated by SAMHD1 T592 phosphorylation. Activation of p53 and the downstream p21 lead to decreased expression of CDK1, the key kinase in SAMHD1 phosphorylation (Cribier *et al*, 2013; Mlcochova *et al*, 2017). As expected from our previous data this loss of CDK1 activity and SAMHD1 phosphorylation was associated with transition of G1-like MDM back into G0 state without induction of apoptosis. In this state, SAMHD1 was dephosphorylated and able to restrict HIV-1.

In summary, we propose a mechanism where ETO-induced DNA damage induces SAMHD1 dephosphorylation via a canonical p53 and p21 pathway in macrophages. As consequence of this regulation, activated dephosphorylated SAMHD1 mediates a block to HIV-1 nuclear import and integration in MDM. ETO and related chemotherapeutic agents are used in HIV infected individuals for malignancies such as lymphoma (Little *et al*, 2003); future work could include exploration of the effect of subsequent DNA damage responses on protection of myeloid target cells from HIV infection. Such insights might assist in design of novel therapeutic interventions, particularly for persistent central nervous system reservoirs that primarily involve macrophages or related myeloid lineage cells.

# Materials and Methods

### Reagents, inhibitors, antibodies and plasmids

Tissue culture media and supplements were obtained from Invitrogen (Paisley, UK), and tissue culture plastic was purchased from TPP (Trasadingen, Switzerland). FCS (FBS) was purchased from Biosera (Boussens, France) and Sigma (Sigma, St. Louis, USA). Human serum from human male AB plasma was of USA origin and sterile-filtered (Sigma). All chemicals, etoposide and campthotecin were purchased from Sigma (St. Louis, MO, USA) unless indicated otherwise. Antibodies used were anti-cdc2 (Cell Signaling Technology, Beverly, USA); anti-CDK2 (H-298, Santa Cruz Biotechnology); anti-pCDK2(Thr160) (Bioss Inc., MA, USA); anti-SAMHD1 (ab67820, Abcam, UK), beta-actin (ab6276, Abcam, UK); mouse anti-MCM2 (BM-28, BD Biosciences, UK) and rabbit anti-MCM2 (SP85) from Sigma; pSAMHD1 ProSci (Poway, CA, USA); p53 (GTX100629, GeneTex, Irvine, CA, USA); p-p53 (9286P, Cell Signaling Technology); H2AX (613301, BioLegend, San Diego, CA, USA); γH2AX (613402, BioLegend); p21(sc-6246, Santa Cruz Biotechnology); p27 (GTX100446, GeneTex); PARP (9542T, Cell Signaling Technology); IRF3 (11904P, Cell Signaling Technology); and 53BP1 (612522, BD Biosciences, UK). pCDNA Vpx Q76A was kind gift from J. Luban (Reinhard *et al*, 2014).

### Cell lines and viruses

293T cells were cultured in DMEM complete (DMEM supplemented with 100 U/ml penicillin, 0.1 mg/ml streptomycin and 10% FCS). SIVmac virus-like particles containing Vpx were prepared as previously described (Goujon *et al*, 2008). VSV-G HIV-1 GFP virus was produced by transfection of 293T with GFP-encoding genome CSGW, packaging plasmid p8.91 and pMDG as previously described (Besnier *et al*, 2002). Stocks of macrophage tropic virus BaL were prepared by infecting a single preparation of MDM. Cell-free supernatants were collected from the infected cultures and stored in liquid nitrogen until further use. Virus stocks were titrated using the focus-forming assay.

HSV-1 KOS K26GFP encoding a VP26-green fluorescent protein (GFP) fusion protein (Blondeau *et al*, 2013). Ad-gfp, adenovirus with E1 and E3 deletion, insertion of CMV-driven gfp in E1 region was produced by Native Antigen Company, Oxfordshire. SFV was kind gift from M. Mazzon (UCL).

### Focus-forming assay

NP-2 CD4/CCR5 cells were plated in 48-well plates at a density of $1.5 \times 10^4$ cells/well, 1 day before infection. Cells were infected for 4 h, washed and cultivated for 60 h in DMEM containing 10% FCS. Cells were fixed in methanol/acetone (1:1) and incubated for 1 h with a mixture of two anti-HIV-1 p24 mAbs (ARP365/366) diluted in PBS/1% FCS, washed in PBS/1% FCS and incubated for 1 h with a secondary anti-mouse β-galactosidase-coupled antibody. After washing with PBS/1% FCS, β-galactosidase substrate solution (0.5 mg/ml 5-bromo-4-chloro-3-indolyl-β-galactoside in PBS containing 3 mM potassium ferricyanide and 1 mM magnesium chloride) was added, and the cells were incubated overnight. Blue-stained infected cell foci were counted microscopically, and virus titres were expressed as ffu/ml.

### Monocyte isolation and differentiation

PBMC were prepared from HIV seronegative donors (after informed consent was obtained), by density-gradient centrifugation (Lymphoprep, Axis-Shield, UK). Monocyte-derived macrophages (MDM) were prepared by adherence with washing of non-adherent cells after 2 h, with subsequent maintenance of adherent cells in RPMI 1640 medium supplemented with 10% human serum and MCSF (10 ng/ml) for 3 days and then differentiated for a further 4 days in RPMI 1640 medium supplemented with 10% fetal calf sera without M-CSF.

### Infection of primary cells using full-length and VSV-G pseudotyped HIV-1 viruses

Macrophage tropic virus BaL was added to MDM and after 4-h incubation removed and cells were washed in culture medium. Cells were fixed in 3% PFA, permeabilized by saponin and stained for intracellular p24 using anti-p24 FITC-conjugated antibody (Santa Cruz Biotechnology, USA). The percentage of infected cells was monitored by flow cytometry using BD FACSCalibur (BD Biosciences, UK) and analysed by CellQuest (BD Biosciences) and FlowJo software (Tree Star, OR, USA). GFP containing VSV-G pseudotyped HIV-1 was added to MDM and after 4-h incubation removed and cells were washed in culture medium. The percentage of infected cells was determined 48 h post-infection by flow cytometry using BD FACSCalibur (BD Biosciences, UK) and analysed by CellQuest (BD Biosciences) and FlowJo software (Tree Star, OR, USA) or by Hermes WiScan automated cell-imaging system (IDEA Bio-Medical Ltd. Rehovot, Israel) and analysed using MetaMorph and ImageJ software.

### Quantitative PCR for total HIV DNA quantitation

Total HIV DNA was detected as previously described (Mlcochova *et al*, 2014). Reverse transcription inhibitor AZT (20 μM) was used as a control to ensure that the total viral DNA measured was the product of productive infection and not a result of DNA contamination of the viral stocks.

### Integrated DNA quantitation

Integrated HIV DNA was measured as described (Liszewski *et al*, 2009). In total, 7,500 cells were assayed per well with five replicates and three independent experiments.

### qPCR for detection of 2-LTR circles

$2 \times 10^5$ MDM were infected with 100 ng of p24 of DNaseI-treated viruses. Cells were washed and harvested for DNA isolation 24 h post-infection. 2-LTR circles were quantified in triplicate by qPCR as described (Apolonia *et al*, 2007). Primers and probe used to detect 2-LTR circles were as follows: 5′-AACTAGAGATCCCTCAGACC CTTTT-3′ and 5′-CTTGTCTTCGTTGGGAGTGAATT-3′, and TaqMan probe 5′-FAM-CTAGAGATTTTCCACACTGAC-TAMRA-3′.

### SDS–PAGE and immunoblots

Cells were lysed in reducing Laemmli SDS sample buffer containing PhosSTOP (Phosphatase Inhibitor Cocktail Tablets, Roche,

Switzerland) at 96°C for 10 min and the proteins separated on NuPAGE® Novex® 4-12% Bis–Tris gels. Subsequently, the proteins were transferred onto PVDF membranes (Millipore, Billerica, MA, USA), the membranes were quenched, and proteins were detected using specific antibodies. Labelled protein bands were detected using Amersham ECL Prime Western Blotting Detection Reagent (GE Healthcare, USA) and Amersham Hyperfilm or AlphaInnotech CCD camera. Protein band intensities were recorded and quantified using AlphaInnotech CCD camera and AlphaView software (Protein Simple, San Jose, California, USA).

### SAMHD1 knock-down by siRNA

$1 \times 10^5$ MDM differentiated in MCSF for 4 days were transfected with 20 pmol of siRNA (L-013950-01, Dharmacon) using Lipofectamine RNAiMAX Transfection Reagent (Invitrogen). Transfection medium was replaced after 18 h with RPMI 1640 medium supplemented with 10% FCS and cells cultured for additional 3 days before infection.

### Quantitative PCR–TaqMan

Total RNA was purified from cell lysates collected in RLT buffer (Qiagen) using the RNEasy Mini kit (Qiagen). cDNA was synthesized using the Omniscript RT Kit (Qiagen), and quantitative (q)PCR of selected genes was performed using the follow ing inventoried TaqMan assays (Applied Biosystems);

CXCL10: forward 5′-TGGCATTCAAGGAGTACCTC-3′ and reverse 5′-TTGTAGCAATGATCTCAACACG-3′. MxA: forward 5′-ATCCTGGG ATTTTGGGGCTT-3′ and reverse 5′-CCGCTTGTCGCTGGTGTCG-3′. ISG56: forward 5′-CCTCCTTGGGTTCGTCTACA-3′ and reverse 5′-G GCTGATATCTGGGTGCCTA-3′. ISG54: forward 5′-CAGCTGAGAA TTGCACTGCAA-3′ and reverse 5′-CGTAGGCTGCTCTCCAAGGA-3′. IRF1: forward 5′-TCTTAGCATCTCGGCTGGACTTC-3′ and reverse 5′-CGATACAAAGCAGGGGAAAAGG-3′. IRF7: forward 5′-GATGTCG TCATAGAGGCTGTTGG-3′ and reverse 5′-TGGTCC TGGTGAAGCT GGAA-3′. STING: forward 5′-TGGGGTGCCTGATAAC-3′ and reverse 5′-TGGCAAACAAAGTCTG-3′. cGAS: forward 5′-GGGAGCCCTGCTG TAACACTTCTTAT-3′ and reverse 5′-TTTGCATGCTTGGGTACAAG GT-3′.

Expression levels of target genes were normalized to glyceraldehyde-3-phosphate dehydrogenase (GAPDH) as previously described (Tsang *et al*, 2009).

### Measurement of deoxynucleoside triphosphate levels in cells

dTTP and dCTP levels were measured essentially as previously described (Arnold *et al*, 2015). Briefly, deoxynucleoside triphosphates were extracted from batches of $1 \times 10^6$ untreated and ETO-treated macrophages according to del Val *et al* (2013). The dNTP levels were quantified by radiolabel incorporation assays performed using oligonucleotide templates detailed in Sherman & Fyfe (1989) and the procedures described in Ferraro *et al* (2010) with the following modifications. Standard curves ranged from 50 to 8,000 fmole, 5 units of Taq polymerase (Invitrogen) was used, and 2.5 μM of α-$^{32}$P-dATP was employed as an incorporation label.

### Immunofluorescence

MDM were fixed in 3% PFA, quenched with 50 mM NH$_4$Cl and permeabilized with 0.1% Triton X-100 in PBS or 90% methanol. After blocking in PBS/1% FCS, MDM were labelled for 1 h with primary antibodies diluted in PBS/1% FCS, washed and labelled again with Alexa Fluor secondary antibodies for 1 h. Cells were washed in PBS/1% FCS and stained with DAPI in PBS for 20 min. Labelled cells were detected using Hermes WiScan automated cell-imaging system (IDEA Bio-Medical Ltd. Rehovot, Israel) and analysed using MetaMorph and ImageJ software.

### Ethics statement

Adult subjects provided written informed consent. Primary Macrophage & Dendritic Cell Cultures from Healthy Volunteer Blood Donors has been reviewed and granted ethical permission by the National Research Ethics Service through The Joint UCL/UCLH Committees on the Ethics of Human Research (Committee Alpha) 2nd of December 2009. Reference number 06/Q0502/92.

**Expanded View** for this article is available online.

## Acknowledgements

This work was funded by a Wellcome Trust Senior Fellowship in Clinical Science to RKG (WT108082AIA). GJT is funded by Wellcome Trust Senior Biomedical Research Fellowship (WT108183), the European Research Council under the European Union's Seventh Framework Programme (FP7/2007-2013)/ERC grant agreement number 339223. IAT is supported by the Francis Crick Institute, which receives its core funding from Cancer Research UK (FC001178), the UK Medical Research Council (FC001178) and the Wellcome Trust (FC001178) and by the Wellcome Trust Senior Investigator Award (108014/Z/15/Z). We would also like to thank Katherine Sutherland, Sarah Watters, Jane Turner and Clare Jolly for helpful advice and reagents.

## Author contributions

PM, RKG and GJT designed experiments; PM, RKG and GJT wrote the manuscript; PM performed experiments; PM, RKG and GJT analysed data. SJC and IAT designed, performed experiments and analysed dNTP data.

## Conflict of interest

The authors declare that they have no conflict of interest.

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
