## [Review Process File · The EMBO Journal]

Manuscript EMBO-2017-96880

DNA damage induced by topoisomerase inhibitors activates SAMHD1 and blocks HIV-1 infection of macrophages

Petra Mlcochova, Sarah J Caswell, Ian A Taylor, Greg J Towers, Ravindra K Gupta

Corresponding author: Ravindra Gupta & Petra Mlcochova, UCL

Review timeline:

Submission date:	09 March 2017
Additional Correspondence	28 April 2017
Editorial Decision:	08 May 2017
Revision received:	14 July 2017
Editorial Decision:	24 August 2017
Revision received:	12 September 2017
Accepted:	22 September 2017

Editor: Karin Dumstrei

Transaction Report:

Additional Correspondence

28 April 2017

Dear Ravi,

Thanks for your patience regarding the review of your manuscript.

I have received 2 reports on your manuscript that I have provided below. I have still not heard back from the third referee and at this stage it is not clear if I will get the last report.

As you can see from the comments, referee #1 appreciates the analysis as is and have no further issues. Referee #2 however is not convinced that the data provides strong enough support for that ETO inhibits HIV infection via SAMHD1. The issues raised are valid and concern key aspects of the paper. They would have to be resolved for consideration here

Before taking a decision on your manuscript I would like to give you the opportunity to respond to the concerns raised by sending me a detailed point-by-point response that also outlines experiments that can be done to address the specific issues raised. Based upon this I will then take the decision.

Don't hesitate to contact me if you have further questions.

1st Editorial Decision

08 May 2017

Thanks for your email. I have now had a chance to take a look at your response and I do find that it makes a good effort to address the concerns raised by referee #2. Please do attempt the experiment suggested by ref #2 - point 2 - to establish causality. This is an important point.

Given this, I would therefore like to invite you to submit a revised version. I should add that it

is EMBO Journal policy to allow only a single major round of revision and that it is therefore important to address the concerns raised at this stage.

When preparing your letter of response to the referees' comments, please bear in mind that this will form part of the Review Process File, and will therefore be available online to the community. For more details on our Transparent Editorial Process, please visit our website: http://emboj.emboPress.org/about#Transparent_Process

Thank you for the opportunity to consider your work for publication. I look forward to your revision.

REFEREE REPORTS

Referee #1:

The authors show here that DNA damage agents etoposide and camptothecin block HIV-1 transduction of macrophages. The drugs do not block transduction by related viruses HIV-2 and SIVsm. The latter two viruses encode a protein, Vpx, that HIV-1 does not encode. Vpx acts to degrade a cellular factor called SAMHD1 that inhibits retroviral reverse transcription and transduction of macrophages. The block to HIV-1 transduction by etoposide occurs after reverse transcription and before integration. It requires the cellular protein SAMHD1. A Vpx mutant that binds to, but does not degrade SAMHD1, rescues HIV-1 from the ETO. SAMHD1 inhibition of HIV-1 is inhibited by phosphorylation. The authors had previously shown that SAMHD1 is dephosphorylated and active in G0 macrophages but phosphorylated and inactive in G1 cells. ETO pushed macrophages into a G0-like state, with activated p53 and p21, loss of CDK1 activity, and thus caused dephosphorylation of SAMHD1, increasing its anti-HIV-1 activity.

This is a very tidy manuscript with convincing data and very interesting results. We have no significant critiques.

Referee #2:

This paper shows that DNA damaging agent ETO inhibits HIV-1 infection in macrophages and that this compound leads to de-phosphorylation of SAMHD1. The finding that ETO inhibits HIV-1 infection in MDM, and not Vpx-containing strains, is interesting and novel. The finding that a Vpx mutant, which does not impact SAMHD1, rescues this effect is interesting. The finding that ETO leads to de-phosphorylation of SAMHD1 is also interesting and novel. However, the link between the two (SAMHD1 as a mediator of the antiviral effect of ETO), which is the key point of the study, is not convincing and essentially correlative. The proposed mechanism is confusing and the data shows inconsistencies, detailed below. A much simpler interpretation of their current data is that ETO inhibits HIV infection independently of SAMHD1, through another mechanism apparently at the level of nuclear import, and that Vpx rescues this effect through another target than SAMHD1.

1. The lack of effect of ETO on RT products would be highly surprising if the mechanism was SAMHD1-dependent. Indeed, they clearly see that SAMHD1 phosphorylation is inhibited by ETO, an event which many labs (including their, ref 21) have shown to be sufficient to alleviate restriction of RT (White et al, CHM 2013; Cribier et al Retrovirology 2013; and later work). In Fig 2D, the inability of SIVVLP to increase the level of RT products is not consistent with the literature (Hrecka et al, Nature 2011), and not consistent with their data confirming an increase in % infected cells (Fig. 2B, 2C). Also, Fig 1F, 2D and 3B (qPCR) lack appropriate controls (RT inhibitors). It is thus not convincing that their experimental method is measuring bona fide RT products. Also, since this is a major point in the study, they would need to show that ETO does not affect RT products (i) over a time-course and (ii) at different MOI.

2. The single experiment that directly tests SAMHD1 is Fig. 1A, where a specific depletion of SAMHD1 is performed. W/o ETO, SAMHD1 KD induces a dramatic increase in viral infection (from few % to >20%), which is expected. Adding ETO on these cells still strongly reduces infection (to few %). In other words, SAMHD1 KD does not at all rescue the effect of ETO. It is

unlikely that this is due to residual levels of SAMHD1 due to partial RNAi, since there is a strong increase of infection in KD cells w/o ETO. Thus, the data is not convincing to show that SAMHD1 depletion "entirely" (line 270), or even partially, abrogates the effect of ETO. To test this hypothesis, they need to perform a dose titration of virus on SAMHD1 KD cells. Can they show that at MOIs that give identical % of infection in control vs SAMHD1 KD cells, ETO has an effect only in control cells, and that it is fully rescued in SAMHD1 KD cells? To strengthen their demonstration, they could also test if the effect of ETO is conserved using cell-cycle arrested U937 cells w/ or w/o reconstituted SAMHD1 (Laguette et al. Nature 2011).

3. Similarly, they need to perform dose-titration of viruses when using Vpx+ SIVVLP to compare similar effective MOIs +/- ETO. In Fig 2B, 2C, it is clear that Vpx increases the effective MOI on macrophages. This is expected since Vpx will degrade SAMHD1, leading to a great increase in RT efficiency. With the current data, they are comparing the effect of ETO between two very different conditions of effective MOI on target cells.

4. They show that Vpx Q67A alleviates the effect of ETO, but there is no impact on SAMHD1 levels or phosphorylation. They suggest that Vpx Q67A would inhibit SAMHD1 catalytic activity by destabilizing the tetramer (l 270-272). If this was what's going on, it would mean that the antiviral effect is mediated by the classical dNTP depletion activity of SAMHD1, leading to RT inhibition, which they fail to detect. Their suggestion is inconsistent with the inability of Vpx Q67A to rescue HIV infection (Fig 3A, also Laguette Nature 2011 and Hrecka Nature 2011). They should measure the impact of Vpx Q67A on SAMHD1 tetramerization. They should measure dNTP levels in macrophages infected +/- Vpx Q67A VLP and +/- ETO and test if indeed they see an increase in dNTP levels. If the dNTP levels do not increase, what is the mechanism that would be mediated by SAMHD1 catalytic activity?

Other points:

- They have mainly used ETO to damage DNA. It's a bit of a stretch to conclude that "DNA damage activates SAMHD1 and blocks infection" as the title claims. They would need to repeat several key experiments with other DNA damaging agents to make such a broad claim.
- The authors mention that the transition from G1-like MDM to G0-like MDMs is not accompanied by apoptosis and base their conclusion on figure 1B. To properly exclude that apoptosis is not induced and how many MDMs actually undergo apoptosis upon ETO treatment, more detailed experiments are necessary.
- In figure 2H the authors calculated fold inhibition of ETO +/- VPX on HIV RT, nuclear entry and integration. This part should be described in more detail in the text.
- Fig 4A: this shows that ETO induces DNA damage. This is a critical point that should be put much earlier in the paper.
- Fig 6 is not very clear. An inhibitory arrow is missing somewhere. Also, the data shows a block of nuclear import, not after nuclear import.
- Abstract and line 67: what is "the association between innate immunity, infection and cancer"? This is vague.
- Line 113: BaL env fuses through CD4-CCR5, but this does not imply that fusion occurs at the plasma membrane. This prevailing view is largely unproven for phagocytic primary or primary-derived cells, and only stems from artificial systems that make use of cancer cell lines over-expressing viral receptors. Please rephrase.
- Line 134-135: Conclude on Vpx, not only Vpr.
- Fig. 1C: Statistical analysis is missing
- Fig. 2C: Only 2 donors, statistical analysis cannot be performed. The experiment must be repeated at least 3 times to perform a statistical analysis.
- Fig. 3A: Stat indicated in legend, but not shown
- Fig. 3B, C, D: This data is not convincing as there are only 2 donors, and no statistical analysis. This experiment needs to be repeated and statistical tests performed.
- Fig 5B, 5C, 5D: How many independent experiments have been performed for these western blot analyses?
- In the main text some parts need to be improved in terms of formulation: Line 156 "...SAMHD1 as it cannot recruit DCAF1 to degrade SAMHD1"; also Line 159 "...SAMHD1 at T592 in the same cells two days later".
- In most figures (1E, 1F, 2A, 2B, 2H, 3A, 4A), an unpaired t-test was used. This is not appropriate, since individual donors were presumably treated in parallel with or without drug. Those are paired

samples, a paired test, and probably more rigorously a non-parametric paired test (it's difficult to assume Gaussian distribution) should be used.

- Scale bars on images are missing

1st Revision - authors' response

14 July 2017

Reviewer 1

This paper shows that DNA damaging agent ETO inhibits HIV-1 infection in macrophages and that this compound leads to de-phosphorylation of SAMHD1. The finding that ETO inhibits HIV-1 infection in MDM, and not Vpx-containing strains, is interesting and novel. The finding that a Vpx mutant, which does not impact SAMHD1, rescues this effect is interesting. The finding that ETO leads to de-phosphorylation of SAMHD1 is also interesting and novel. However, the link between the two (SAMHD1 as a mediator of the antiviral effect of ETO), which is the key point of the study, is not convincing and essentially correlative. The proposed mechanism is confusing and the data shows inconsistencies, detailed below. A much simpler interpretation of their current data is that ETO inhibits HIV infection independently of SAMHD1, through another mechanism apparently at the level of nuclear import, and that Vpx rescues this effect through another target than SAMHD1.

1. *The lack of effect of ETO on RT products would be highly surprising if the mechanism was SAMHD1-dependent. Indeed, they clearly see that SAMHD1 phosphorylation is inhibited by ETO, an event which many labs (including their, ref 21) have shown to be sufficient to alleviate restriction of RT (White et al, CHM 2013; Cribier et al Retrovirology 2013; and later work).*

RESPONSE: The effect of ETO on RT is indeed surprising. SAMHD1 KD depletion illustrates that ETO mediated reduction of HIV infection is almost entirely SAMHD1-dependent (Figure 2A, 2B). ETO treatment triggers a return of MDM to a G0 state that is accompanied by SAMHD1 dephosphorylation and dNTP loss as would be expected in an arrested cell state (Figs 2, EV1, 3). Our approach uses direct manipulation of endogenous SAMHD1 phosphorylation in human macrophages - a highly original experimental set-up. Regarding the lack of effect on HIV DNA synthesis it is possible that ETO has effects on other cellular components/processes that maintain RT efficiency in the face of lower total cellular dNTP levels. We discuss this possibility (lines 272-290).

In addition, the studies cited by the reviewer do not conclusively support an effect of SAMHD1 phosphorylation on HIV DNA synthesis in primary macrophages. White et.al. and Cribier et.al. measured infection and noted SAMHD1 phosphorylation was associated with increased infection. These studies did not measure HIV DNA synthesis. In regards to our study, we showed that whilst RT increased modestly (3x) under FCS culture (associated with phosphorylation of SAMHD1), it did not fully explain the increase in infection – see Fig 1A and 1B from Mlcochova et al EMBO J 2017). This would be consistent with a post RT effect. In addition, a number of studies have reported that restriction of HIV is regulated by phosphorylation at T592 via dNTP **hydrolase independent** mechanisms (Welbourne and Strebel 2016; Welbourne et al 2013; Bhattacharya et al 2016). Thus significant evidence points to an DNA synthesis independent block regulated by SAMHD1 phosphorylation.

In Fig 2D (now Fig.2F), the inability of SIV VLP to increase the level of RT products is not consistent with the literature (Hrecka et al, Nature 2011), and not consistent with their data confirming an increase in % infected cells (Fig. 2B, 2C).

RESPONSE: Experiments in figure 2 were performed in the presence of FCS culture, where SAMHD1 is phosphorylated and inactive (Mlcochova et al, 2017). Therefore one would not expect an increase in RT products with SIV VLP. We have now clarified in the text that FCS culture was used (Line 154). Therefore our data are not inconsistent with previous publications. Moreover, Fig. 2B, 2C shows only 1.6 fold and 1.5 fold increase (see below) in % infected cells after SIV VLP addition, which is not statistically significant.

Also, Fig 1F, 2D and 3B (qPCR) lack appropriate controls (RT inhibitors). It is thus not convincing that their experimental method is measuring bona fide RT products. Also, since this is a major point in the study, they would need to show that ETO does not affect RT products (i) over a time-course and (ii) at different MOI.

We are certain that our assay is measuring HIV RT products but the reviewer did not appreciate that SAMHD1 is phosphorylated/inactive in the experiments cited here. We have clarified in the text that FCS culture was used (line 154). In addition, we have repeated, as suggested, the experiments and treated MDM with the reverse-transcription inhibitor AZT as a control in Figures 1G, 2F, 3D. These experiments confirm the specificity of our qPCR method.

Furthermore, we have included new experiments to confirm that ETO does not affect RT products (i) over a time-course and (ii) at different MOI. These are now Fig. 1J-L in the revised manuscript.

2. The single experiment that directly tests SAMHD1 is Fig. 1A, where a specific depletion of SAMHD1 is performed. W/o ETO, SAMHD1 KD induces a dramatic increase in viral infection (from few % to >20%), which is expected. Adding ETO on these cells still strongly reduces infection (to few %). In other words, SAMHD1 KD does not at all rescue the effect of ETO. It is unlikely that this is due to residual levels of SAMHD1 due to partial RNAi, since there is a strong increase of infection in KD cells w/o ETO. Thus, the data is not convincing to show that SAMHD1 depletion "entirely" (line 270), or even partially, abrogates the effect of ETO. To test this hypothesis, they need to perform a dose titration of virus on SAMHD1 KD cells. Can they show that at MOIs that give identical % of infection in control vs SAMHD1 KD cells, ETO has an effect only in control cells, and that it is fully rescued in SAMHD1 KD cells?

We disagree with this reviewer's interpretation. We believe the reviewer is referring to figure 2A, not 1A. In 2A SAMHD1 knockdown increases infection by 3 fold, a small increase, versus 84 fold for on ETO treatment a large increase (see below). Therefore there is a very large difference in sensitivity to ETO when SAMHD1 is present. Although we do not think it is critical for our conclusions we nonetheless performed the suggested experiment that is now included in the revised version as Fig. 2D. These data show that the effect of ETO is independent of viral dose and is rescued at all HIV doses by SAMHD1 depletion.

To strengthen their demonstration, they could also test if the effect of ETO is conserved using cell-cycle arrested U937 cells w/ or w/o reconstituted SAMHD1 (Laguette et al. Nature 2011).

We have not used cell lines (such as U937 or THP-1) for our ETO experiments as we found ETO to be toxic in these cells consistent with the well described toxicity of ETO in cancer cells and its use as a cancer treatment.

Similarly, they need to perform dose-titration of viruses when using Vpx+ SIVVLP to compare similar effective MOIs +/- ETO. In Fig 2B, 2C, it is clear that Vpx increases the effective MOI on macrophages. This is expected since Vpx will degrade SAMHD1, leading to a great increase in RT efficiency. With the current data, they are comparing the effect of ETO between two very different conditions of effective MOI on target cells.

These suggested experiments and they are now included in the revised version as Fig.2E. As before viral dose does not impact experimental outcome.

2. They show that Vpx Q67A alleviates the effect of ETO, but there is no impact on SAMHD1 levels or phosphorylation. They suggest that Vpx Q67A would inhibit SAMHD1 catalytic activity by destabilizing the tetramer (1 270-272). If this was what's going on, it would mean that the antiviral effect is mediated by the classical dNTP depletion activity of SAMHD1, leading to RT inhibition, which they fail to detect. Their suggestion is inconsistent with the inability of Vpx Q67A to rescue HIV infection (Fig 3A, also Laguette Nature 2011 and Hrecka Nature 2011). They should measure the impact of Vpx Q67A on SAMHD1 tetramerization.

Our data are consistent with Laguette Nature 2011 and Hrecka Nature 2011 showing that Vpx Q76A cannot rescue HIV infection in MDM under human serum conditions (see Figure below) where the cells have low dNTPs (1a, 1b) or when cultured in FCS, leading to high dNTPs (condition used throughout this manuscript) (2a, 2b). Evidently macrophages treated with ETO are responding differently and, unlike in untreated cells, Vpx Q76A is able to rescue HIV infection (1c,d; 2c,d). Figures below are now included in revised version of manuscript as Fig. 3A-B.

legend: black bars represent human serum conditions and white bars FCS conditions

Differences in tetramerisation as an explanation are only one possibility amongst many. When we referred to tetramerisation as an explanation we were merely speculating about the mechanism (this was in the discussion).

3. *They should measure dNTP levels in macrophages infected +/- Vpx Q67A VLP and +/- ETO and test if indeed they see an increase in dNTP levels. If the dNTP levels do not increase, what is the mechanism that would be mediated by SAMHD1 catalytic activity?*

We measured dTTP and dCTP levels in MDM +/- ETO and +/- SIV VLP Q76A. dNTP levels did not increase with infection with SIV VLP Q76A in MDM (Fig EV3), similar to a result previously obtained in dendritic cells (Reinhard, C., Retrovirology 2014). Nevertheless, ETO treatment does decrease dNTP levels in MDM (Fig EV1). These experiments present a working model to explore whether SAMHD1 restriction of RT can be dNTP-independent. We speculate that SAMHD1 may bind viral DNA directly or indirectly and therefore mediate post RT restriction. Indeed there are multiple reports indicating a dNTP hydrolase independent mechanism for SAMHD1 mediated restriction of HIV-1. We have included this in the discussion (lines 267-290 and 292-300).

Other points:

- *They have mainly used ETO to damage DNA. It's a bit of a stretch to conclude that "DNA damage activates SAMHD1 and blocks infection" as the title claims. They would need to repeat several key experiments with other DNA damaging agents to make such a broad claim.*

We agree. We have changed the title to '**DNA damage induced by topoisomerase inhibitors activates SAMHD1 and blocks HIV-1 infection of macrophages**'.

- *The authors mention that the transition from G1-like MDM to G0-like MDMs is not accompanied by apoptosis and base their conclusion on figure 1B. To properly exclude that apoptosis is not induced and how many MDMs actually undergo apoptosis upon ETO treatment, more detailed experiments are necessary.*

Cleavage of PARP-1 by caspases is accepted as a distinctive feature of apoptosis [Kaufmann et al., Casiano et al.]. We argue that measurement of PARP cleavage in Figure 5 combined with the lack of cell death evidenced in Figure 1C is excellent evidence to conclude a lack of apoptosis and cell death after DNA damage in macrophages. We have nonetheless rephrased the relevant sentence in line 236 – 238.

Kaufmann SH, Desnoyers S, Ottaviano Y, Davidson NE, Poirier GG. Specific proteolytic cleavage of poly(ADP-ribose) polymerase: an early marker of chemotherapy-induced apoptosis. *Cancer Res.* 1993 Sep 1;53(17):3976-85. PubMed PMID: 8358726.

Casiano CA, Martin SJ, Green DR, Tan EM. Selective cleavage of nuclear autoantigens during CD95 (Fas/APO-1)-mediated T cell apoptosis. *J Exp Med*. 1996 Aug 1;184(2):765-70. PubMed PMID: 8760832; PubMed Central PMCID: PMC2192733.

- *In figure 2H the authors calculated fold inhibition of ETO +/- VPX on HIV RT, nuclear entry and integration. This part should be described in more detail in the text.*

We agree. We have modified Figure 2 and removed panel 2H to simplify the message of this finding.

- *Fig 4A: this shows that ETO induces DNA damage. This is a critical point that should be put much earlier in the paper.*

We agree. Panel 4A has been moved to Figure 1 as Fig.1A.

- *Fig 6 is not very clear. An inhibitory arrow is missing somewhere. Also, the data shows a block of nuclear import, not after nuclear import.*

We agree. We have changed this Figure and clarified the legend. Our data could be interpreted as block to nuclear import. However, we are measuring 2-LTR circles and integrated proviruses. Whilst such an observation is often described as a block to nuclear entry we are keen to point out that these data are also consistent with direct inhibition of circle and provirus formation. This is important because the molecule having the effect (SAMHD1) is clearly in the nucleus.

- *Abstract and line 67: what is "the association between innate immunity, infection and cancer"?* This is vague.

RESPONSE: We agree with the reviewer and have deleted this sentence.

- *Line 113: BaL env fuses through CD4-CCR5, but this does not imply that fusion occurs at the plasma membrane. This prevailing view is largely unproven for phagocytic primary or primary-derived cells, and only stems from artificial systems that make use of cancer cell lines over-expressing viral receptors. Please rephrase.*

RESPONSE: We agree with reviewer and modified this.

- *Line 134-135: Conclude on Vpx, not only Vpr.*

RESPONSE: We have modified the text accordingly – we have shown early in the paper that Vpx abrogates the impact of ETO and therefore do not feel it is important to repeat this information later in the paper where we specifically look at Vpr.

- *Fig. 1C: Statistical analysis is missing*

RESPONSE: Figure 1 has been modified. Fig.1C is now 1D. We have added statistical analysis and immunoblots showing SAMHD1 expression.

- *Fig. 2C: Only 2 donors, statistical analysis cannot be performed. The experiment must be repeated at least 3 times to perform a statistical analysis.*

RESPONSE: We apologize; we have wrongly stated that these results are from 2 donors. 2C graph actually represent average of 4 independent experiments and immunoblot is a representative example.

- *Fig. 3A: Stat indicated in legend, but not shown*

RESPONSE: Figure 3 has been changed and all panels contain statistical analysis now.

- Fig. 3B, C, D: *This data is not convincing as there are only 2 donors, and no statistical analysis. This experiment needs to be repeated and statistical tests performed.*

RESPONSE: We have changed Figure 3. Fig.3B-D is now Fig.3 C-F. We have repeated the experiments and performed statistical analysis. We have also added new panel 3E showing changes to 2-LTR circles to complete our data.

- Fig 5B, 5C, 5D: *How many independent experiments have been performed for these western blot analyses?*

RESPONSE: These experiments have been done 3 times. We are using a representative example in Figure 5 B-D. The figure legend has been changed accordingly to clarify this.

- *In the main text some parts need to be improved in terms of formulation: Line 156 "...SAMHD1 as it cannot recruit DCAF1 to degrade SAMHD1"; also Line 159 "...SAMHD1 at T592 in the same cells two days later".*

RESPONSE: We have rephrased the formulations.

- *In most figures (1E, 1F, 2A, 2B, 2H, 3A, 4A), an unpaired t-test was used. This is not appropriate, since individual donors were presumably treated in parallel with or without drug. Those are paired samples, a paired test, and probably more rigorously a non-parametric paired test (it's difficult to assume Gaussian distribution) should be used.*

- *Scale bars on images are missing*

RESPONSE: We agree with reviewer. We have used a paired t-test in revised version. Scale bars were added to Figures 1A and 4B.

2nd Editorial Decision

24 August 2017

Thank you for submitting your revised manuscript to The EMBO Journal. Your revision has now been re-reviewed by referee #2 and the comments are provided below.

As you can see, the referee appreciates the introduced revisions and finds that the analysis has been improved. There is however still one outstanding issue - the referee is not convinced that the analysis shows strong enough support for that ETO-induced restriction is mediated via SAMHD1. This is an important issue that should be sorted out and I am open to consider a revised version that addresses this concern.

The referee suggests a few different approaches to address this concern. Take a look at the suggestions and let's discuss further.

REFEREE REPORT

Referee #2:

The authors have improved the manuscript. The study is clear, well performed and interesting. The restrictive phenotype induced by ETO and alleviated by Vpx is very exciting and novel. Unfortunately the central demonstration that ETO-induced restriction is mediated by SAMHD1 remains insufficiently convincing. Currently, the data is rather consistent with the existence of a second target for Vpx.

As suggested by the Reviewer, they performed a dose titration (Fig 2D). However, Fig 2D does not show a clear titration (even considering that this is a log scale). The critic of the Reviewer remains: "Can they show that at MOIs that give identical % of infection in control vs SAMHD1 KD cells, ETO has an effect only in control cells, and that it is fully rescued in SAMHD1 KD cells?". This critical question cannot be addressed with the data of Fig 2D. They need to perform a more extensive titration.

This demonstration is the central fulcrum of the study, because the current data is rather inconsistent with what is known about SAMHD1 restriction: here, (1) RT is not affected, but 2 LTR circles and viral integration are affected; (2) SAMHD1 dephosphorylation is unlikely mediating the restrictive mechanism induced by ETO (1.328), because they show that Vpx Q67A rescues infection despite an intact ETO-induced SAMHD1 dephosphorylation; (3) Vpx Q67A rescues the effect of ETO but has no impact on SAMHD1 that they can detect (4) SAMHD1 reconstitution experiments could not be performed. They evoke a 'second mechanism of SAMHD1 restriction' (1.268-270). The papers cited make use of SAMHD1-reconstituted cell lines, as previously suggested by the Reviewer. I understand their point that ETO is toxic for cell lines. As an alternative, they can generate macrophages from SAMHD1 KO vs WT mice (previously published by the authors: Mlcochova 2017), and infect the BMDM with a dose-titration of VSV-G pseudotyped HIV-1 +/- ETO treatment.

A complementary approach to demonstrate the implication of SAMHD1 would be to identify a Vpx mutant that does not bind SAMHD1, but still binds DCAF1 and localizes to the nucleus (for example), and to show that such mutant would be unable to rescue the ETO-induced restriction. Vpx-SAMHD1 structures have been solved in the beautiful work of Ian Taylor. Thus, identification of such Vpx mutant should be feasible. Alternately, although less direct, they could test a panel of Vpx alleles (Lim ES, CHM 2012) and determine if there is a match between the ability of Vpx alleles to degrade human SAMHD1 and the ability to rescue the ETO-induced restriction.

Minor point:

The authors emphasize in their rebuttal that in FCS culture condition, SAMHD1 is phosphorylated and inactive in MDM. It is not clear if this view has reached a consensus in the field. For example, the seminal study of Hrecka et al. Nature 2011 used bovine serum and reported a strong SAMHD1 restriction alleviated by Vpx.

2nd Revision - authors' response

12 September 2017

The authors have improved the manuscript. The study is clear, well performed and interesting. The restrictive phenotype induced by ETO and alleviated by Vpx is very exciting and novel. Unfortunately the central demonstration that ETO-induced restriction is mediated by SAMHD1 remains insufficiently convincing. Currently, the data is rather consistent with the existence of a second target for Vpx.

As suggested by the Reviewer, they performed a dose titration (Fig 2D). However, Fig 2D does not show a clear titration (even considering that this is a log scale). The critic of the Reviewer remains: "Can they show that at MOIs that give identical % of infection in control vs SAMHD1 KD cells, ETO has an effect only in control cells, and that it is fully rescued in SAMHD1 KD cells?". This critical question cannot be addressed with the data of Fig 2D. They need to perform a more extensive titration.

We have performed the experiment suggested by the reviewer where we obtain identical % infection in control and SAMHD1 KD cells by using different MOIs. We found that ETO had a potent inhibitory effect in control cells with SAMHD1 but no significant effect in SAMHD1 KD cells (Figure EV3C). This experiment was conducted in 3 different donors and represents average between donors.

We are extremely pleased with this result and are confident that this represents sufficient proof that effect of ETO on HIV-1 infection is SAMHD1 dependent.

This demonstration is the central fulcrum of the study, because the current data is rather inconsistent with what is known about SAMHD1 restriction: here, (1) RT is not affected, but 2 LTR circles and viral integration are affected; (2) SAMHD1 dephosphorylation is unlikely mediating the restrictive mechanism induced by ETO (1.328), because they show that Vpx Q67A rescues infection despite an intact ETO-induced SAMHD1 dephosphorylation; (3) Vpx Q67A rescues the effect of ETO but has no impact on SAMHD1 that they can detect (4) SAMHD1 reconstitution experiments could not be performed. They evoke a 'second mechanism of SAMHD1 restriction' (1.268-270). The papers cited make use of SAMHD1-reconstituted cell lines, as previously suggested by the Reviewer. I understand their point that ETO is toxic for cell lines. As an alternative, they can generate macrophages from SAMHD1 KO vs WT mice (previously published by the authors: Mlcochova 2017), and infect the BMDM with a dose-titration of VSV-G pseudotyped HIV-1 +/- ETO treatment.

Recently published work of Daddacha et.al. (Cell Reports 2017) shows new dNTP-independent function of SAMHD1 in DNA end resection. This study shows that SAMHD1 binds another protein during DNA double-strand breaks in response to DNA damage and promotes homologous recombination. This suggest that SAMHD1 indeed has more functions than only dNTP regulation and it is feasible that this new function may play role in HIV-1 integration that is recognized in cells as a DNA damage.

The approach using mouse BMDM would be highly elegant; unfortunately the main problem with using BMDM is that these cells still divide in culture. As ETO is inducing block in G2/M in dividing cells the effect of ETO in BMDM is very different to the effect we are measuring in human macrophages (transition from G1-IIke to G0 phase).

A complementary approach to demonstrate the implication of SAMHD1 would be to identify a Vpx mutant that does not bind SAMHD1, but still binds DCAF1 and localizes to the nucleus (for example), and to show that such mutant would be unable to rescue the ETO-induced restriction. Vpx-SAMHD1 structures have been solved in the beautiful work of Ian Taylor. Thus, identification of such Vpx mutant should be feasible. Alternately, although less direct, they could test a panel of Vpx alleles (Lim ES, CHM 2012) and determine if there is a match between the ability of Vpx alleles to degrade human SAMHD1 and the ability to rescue the ETO-induced restriction.

We think that our first experiment is sufficient to demonstrate that effect of ETO on HIV-1 infection is SAMHD1 dependent. Mutant experiments can be difficult to interpret and represent only indirect evidence.

Minor point:

The authors emphasize in their rebuttal that in FCS culture condition, SAMHD1 is phosphorylated and inactive in MDM. It is not clear if this view has reached a consensus in the field. For example, the seminal study of Hrecka et al. Nature 2011 used bovine serum and reported a strong SAMHD1 restriction alleviated by Vpx.

Our study from earlier this year has been positively received by the international scientific community. In the pivotal study by Hrecka et.al. (Nature 2011) human MDM were cultured in FBS. However, their protocol had a number of variations from ours which could have led to their observations. The Hrecka et.al. study did not measure phosphorylated SAMHD1 and therefore we cannot be certain as to the explanation. It is clear that FBS culture using our method (a widely used protocol) leads to phosphorylation of SAMHD1 and loss of antiviral activity.

Corresponding Author Name: RAVINDRA K GUPTA

Journal Submitted to: EMBO J

Manuscript Number: EMBOJ-2017-96880